# The Optimal Concentration of Nanoclay Hydrotalcite for Recovery of Reactive and Direct Textile Colorants

**DOI:** 10.3390/ijms23179671

**Published:** 2022-08-26

**Authors:** Daniel López-Rodríguez, Bàrbara Micó-Vicent, Marilés Bonet-Aracil, Francisco Cases, Eva Bou-Belda

**Affiliations:** 1Departamento de Ingeniería Textil y Papelera, Universitat Politècnica de València, Plaza Ferrándiz y Carbonell s/n, CP 03801 Alcoy, Spain; 2Departamento de Ingeniería Gráfica, Universitat Politècnica de València Plaza Ferrándiz y Carbonell s/n, CP 03801 Alcoy, Spain

**Keywords:** nanoclay, dye adsorption, nanoclay pigment, direct dye recovery, removal of dyes, reactive dye recovery, hydrotalcite, hybrid pigments, natural dyes, biopolymer

## Abstract

Concerns about the health of the planet have grown dramatically, and the dyeing sector of the textile industry is one of the most polluting of all industries. Nanoclays can clean dyeing wastewater using their adsorption capacities. In this study, as a new finding, it was possible to analyze and quantify the amount of metal ions substituted by anionic dyes when adsorbed, and to determine the optimal amount of nanoclay to be used to adsorb all the dye. The tests demonstrated the specific amount of nanoclay that must be used and how to optimize the subsequent processes of separation and processing of the nanoclay. Hydrotalcite was used as the adsorbent material. Direct dyes were used in this research. X-ray diffraction (XRD) patterns allowed the shape recovery of the hydrotalcite to be checked and confirmed the adsorption of the dyes. An FTIR analysis was used to check the presence of characteristic groups of the dyes in the resulting hybrids. The thermogravimetric (TGA) tests corroborated the dye adsorption and the thermal fastness improvement. Total solar reflectance (TSR) showed increased radiation protection for UV-VIS-NIR. Through the work carried out, it has been possible to establish the maximum adsorption point of hydrotalcite.

## 1. Introduction

The planet’s resources are limited and, nowadays, managing and conserving them is of global importance. Therefore, all possible measures are being taken to make the most of these resources and not waste them. The textile chemical industry is the sector that produces the most wastewater with the greatest chemical activity on the Earth [1]. The recycling of industrial wastewater is perceived as an ever-growing need. The concentration of dyes in effluents has been reported to be around 50–1000 ppm, although cases of lower than 10–50 ppm have been found [2].

Textile dye materials are classified into different categories depending on the dye material used, such as basic [3,4], acidic, direct, reactive, vat, and disperse dyes. Direct and reactive dye materials, in particular, are commonly produced in large amounts for textile industry printing and dyeing processes to bring color to cellulosic materials, due to their availability and range of colors [5]. Due to their toxicity, these dyes have both been extensively studied for their removal from wastewater by various techniques [6,7,8,9].

In an effort to recover all the natural resources that we enjoy, the use of nanocomposites and nanoclays has become more common in industry. When small amounts of nanoclays are incorporated into polymeric matrices, the result is called a nanocomposite [10]. Previous works have demonstrated the efficacy of this type of element and have achieved successes in corresponding trials [11,12,13,14,15,16,17,18,19,20]. The Awual group has reported a growing trend to use composite materials for the removal of different dyes, based on the specific surface area of the adsorbing agent and functionality [21,22,23].

It has been shown that nanoclays can be used in industry as effective absorbents for different chemical compounds, and therefore, they have been used in the purification of textile wastewater and many other industries. Thanks to their specific characteristics, they have the ability to form hybrids by absorbing and retaining residual industrial dyes that have not been absorbed during the dyeing process. These hybrids can be eliminated after filtering or even given new uses, thus, making full use of all the materials [24,25,26]. The properties of nanoclay allow novel applications to be defined, including applications which significantly reduce the amount of colorants that reach wastewater. This has led, in turn, to new lines of research into the possibility of reusing colorants that are not being used and are going to waste. In addition, nanoclays are an abundant and low-cost material [27,28], which makes them very cost-effective considering the cost-effectiveness ratio. Consequently, hydrotalcite was chosen for this research in view of the results of other works that have demonstrated the adsorption capacity of anionic textile dyes [24,29,30,31,32,33].

Hydrotalcite (HC) is a mineral that is classified as a nanoclay because its dimensions are less than 20 nm. This laminar nanoclay is of interest to the scientific community due to the different possible applications that it offers, such as in medicine, sludge cleaning, and petrochemical spills [34]. Hydrotalcite has a huge capacity to adsorb negatively charged compounds and this adsorption can be carried out using different procedures [32,35]. The most simple and effective procedure is based on the direct adsorption of compounds that are in dispersion, although there are certain limitations due to the crystallinity that the solid may present, the temperature, the pH, the size of the anions, etc. [30,36,37]. It has been shown that hydrotalcite has shape memory, which allows it to be calcined at 600 °C, and therefore, once it is in contact with an aqueous solution it recovers its shape. In this initial structure recovery process, the anions present in the solution are incorporated into the structure recovered by HC [24,25,26]. An alternative method involves a synthetic process in which the nanoclay incorporates a specific anion into the structure of the hydrotalcite. This method is called co-precipitation [38].

For this work, direct dyes and a reactive dye were chosen, with the aim of carrying out a study on the adsorption of the most common dyes used in cotton dyeing, since these have a loss in effluents of up to 30% [39]. In addition, when studying direct and reactive dyes, a statistical experimental design was followed to consider the influence of molecule size on the adsorption phase. Within the range of selected colorants, complete trichromy was also used, which formed part of this team’s plans for future reuse work.

## 2. Results

### 2.1. Final Concentrations in Solution

The results obtained confirm that the nanoclay has a significant capacity to adsorb dyes (Table 1), as has been shown in other studies [40,41,42,43,44] that tested the adsorption properties of HC for removing several dyes such Congo red, acid red 1, methyl orange, and methylene blue. The tests on Samples 13–16 were completed and it was observed that the nanoclay was totally saturated and not capable of absorbing more dye. After reaching the saturation point of the nanoclay, it is possible to define the relationship “x” defined above, which is the scientific contribution of this work. Prior to these analyses, dye adsorption tests were carried out on several real wastewaters collected from a textile dyeing industry, thus, confirming the adsorption capacity and making it possible to extrapolate the results of this work to the real industry.

The adsorption for each dye shows the hybrid pigments process performance. For this reason, the total amount of dye loaded into the nanoclay as Ads(%) was used as the response variable for the DoE analysis. Table 2 shows that both of the selected factors, i.e., the dye structure and the nanoclay/dye ratio (x), are significant in the cleaning or hybrid synthesis process that it is studied. The AB interaction is not significant, which means that the trends observed in the main factors are the same as that observed in the interaction (Figure 1). There were no different behaviors observed depending on the dye structure related to the dye/clay concentration. 

As it can be observed in Figure 1, the Direct Blue 71 Sample 3, with the highest molecular weight, has less significant adsorption than the other three dyes. On the one hand, the differences are lower than 2% due to the experimental conditions. On the other hand, the differences between the nanoclay/dye concentrations show that the lowest x ratio, means more adsorbed dye, as expected, because as the dye concentration increased the adsorption grew. However, there are no significant differences between the 0.5 and 2 x levels and the 100 and 200 concentration levels. 

### 2.2. Color Measurements

The color measurements of the dye-clay hybrids represented in the chromatic diagrams are shown below. For the colorimetric calculations of each hybrid pigment, the reflectance (λ) measurements were used. The main CIE 15:2004 standard [45] guidelines of the International Commission on Illumination and Color (CIE) were followed to make the absolute and relative colorimetric comparisons. The colorimetric CIELAB parameters codified by the CIE 1931 XYZ patron and D65 standard illumination were used. The CIE a*b* and CIE-Cab*L* diagrams in Figure 2 show that there are changes in the hue due to the dye and the concentration. For example, the samples with Yellow Drimaren are more yellowish at low concentrations (x = 2 ore 0.5) than the samples at high concentrations (x = 200 and 100). The same phenomenon can be found in the samples with Direct Red 23. 

However, the sample with x = 2 with the Direct Red 23, Direct Blue 71, and Direct Blue 199 are apparently achromatic as compared with the less concentrated samples at x = 0.05 with the same dyes. The chroma is affected by the dye concentration and the hue. With yellow samples, it is more difficult to get darker and achromatic samples, as we can see in the L*-Cab* highlighted areas. In addition, as the dye ratio increases (x decrease), the samples L* is low and the chroma decrease, allowing a wider range of colors to be covered. There is a hue shift towards green in the Direct Blue 199 samples when the concentrations increase to x = 100 and 200. The color range represented by the area under the L* C_ab_* diagrams is bigger with the samples with the Direct Blue 199 than the samples with the Direct Blue 71 which are less chromatic and less bright. Finally, the color range with the samples with Direct Red 23 is wider in the samples with Direct Blue 71. 

### 2.3. Total Solar Reflectance (TSR)

The amount of energy that is absorbed by solar radiation in the most superficial layers of a substrate is what determines the amount of heat accumulated on that surface. One of the most determining factors for this phenomenon is the exposure time that the surface undergoes. To achieve coatings with cold surfaces, the maximum reflection of solar radiation must be achieved. This reflection capacity of a body is expressed numerically by total solar reflectance (TSR), considering 0% total absorption and 100% complete reflection.

To calculate solar reflectance, it is necessary to have the data of the solar reflectance of the raw substrate and apply the solar weighting factors for each wavelength to be analyzed. All these calculations are included in the ASTM G173-03 standards [46]. To calculate the degree of total solar absorption performed by the hybrid, (1-TSR) must be performed.

Table 3 shows a comparison from maximum (x = 0.5) to minimum (x = 200) saturation of the nanoclay after adsorption of the dye. Figure 3 represents the highest and lowest x ratio for the 4 dyes. In all these samples the visible light behavior depends on the dye structure affecting the color perception (hue, chroma and lightness). In the near infrared from 700–1400 nm the differences due to the x ratio are significant, however, from 1400–2400 nm the behavior of the samples is closer. Here the nanoclay structure has a stronger effect on the TSR (%) contribution than the dye concentration. 

In Figure 4, the Direct Red 21 dye and all its hybrids are studied to assess how this change in concentration affects the results. As can be observed, the dye concentration affects the hybrid pigment reflectances and dye ratio increases (lower × values), the reflectance factor 𝜌 (%) decreases over all the spectra measured. The differences in the visible light between the x = 2 and x = 0.05 samples seems low, but those differences increase significantly in the near infrared from 700–2400 nm affecting the final TSR (%) value. This means that hybrid pigments with similar color perception, can obtain different TSR (%) values affecting the future applications that should be considered. 

Figure 5, shows TSR (%) values seem to be correlated with the nanoclay/dye concentration. As can be expected, the samples with low L* values have low TSR (%) due to the lower reflectance mainly in the visible spectrum. In addition, yellow samples have more TSR (%) for the same reasons. However, more differences can be found between all the dye structures and x ratios but is not possible to know if those differences are stastically significant without the variance analysis. 

The x ratio and the dye structure affects the TSR (%) value in the hybrid pigments, and there are no interactions between these factors as can be found in the ANOVA analysis (Table 4). The Yellow drimaren pigments has high TSR (%) values, meanwhile there are no significant differences between the blues and the reds. Also dye concentration increases, the TSR (%) decreases but there are no significant differences between x = 2 and x = 0.5, or x = 100 and x = 200. To achieve a high TSR value x = 100 is sufficient, and it is not a problem to increase the dye ratio from x = 2 to x = 0.5. The TSR (%) is the same and the color increases significantly as can be seen in Figure 6. 

### 2.4. Thermogravimetry (TGA)

The results obtained in the TGA are shown in Figure 7, Figure 8, Figure 9 and Figure 10. Each figure shows the degradation vs. temperature curves of each colorant (B199, B71, R23, and YD) and the degradation of the nanoclay (HC) is also analyzed. The curves at the bottom of each graph show the lines d(YD), d(B71), d(R23), d(B199), and d(HC), which are derived from the first curves indicating specific degradation peaks (DTGA). The results show that, on the one hand, Direct Red 23 and Reactive Yellow start to degrade earlier, but their decline is gradual and less marked within a range of approximately 217–556 °C. On the other hand, Blue 199 shows a marked peak at 367–482 °C. The degradation of Direct Blue 71 is the lowest and its degradation is the slowest. 

Figure 8 shows a comparison of the TGA and DTGA of HC and Sample 3. As can be seen, hydrotalcite suffers a loss of water including physiosorbed and interlaminar in the range of 100–170 °C. The second loss corresponding to the range of 170–280 °C is attributed to the loss of lamellar OH groups. The third loss, recorded between 280 and 600 °C, can be assigned to the loss of carbonate ions and combustion of the molecule fragment [47,48,49]. When hydrotalcite is analyzed alone without having previously adsorbed any dye, there are peaks at 208, 297, and 411 °C. The next step was to analyze the hybrid pigments after adsorption to see how the nanoclay affects thermal behavior. Figure 7, Figure 8, Figure 9 and Figure 10 show how the initial properties of the dyes have been notably altered. 

Figure 7 shows the thermal behavior of B199. Two peaks are observed, one at 411 °C and a very prominent one at 456 °C. After adsorption on HT, the height of the original peak of the nanoclay slightly decreases in the range between 164 and 208 °C.Peaks at 297 and 411 °C are also observed, although with less loss of mass at these points. For Sample 11, two new peaks can be seen at 432 and 494 °C, most likely due to the considerable loss of mass suffered by the dye at 456 °C which is manifested in these two new peaks once adsorbed by HT.

In the case of the dye B71 (Figure 8), several peaks at around 480, 562, 625, and 724 °C are observed, which disappear after adsorption on HC, although a very mild peak is seen at 639 °C, most likely produced by the gradual degradation of the colorant between the 562 and 625 °C, which has been softened by the effect of the nanoclay. Moreover, peaks at 195, 293, and 373 °C also appear, typical of nanoclay, as seen in Figure 8 and Figure 10 marking the peaks at 208, 297, and 411 °C. 

In the hybrid formed by the Red 23 dye and the nanoclay (Figure 9), there are peaks at 447, 501, and 799 °C which completely disappear when the dye is adsorbed by the nanoclay. A single, wide peak is observed around 376 °C, which is also observed in hydrotalcite.

Finally, with Yellow DR (Figure 10), two peaks can be observed at 430 and 532 °C, which completely disappear when the hybrid is formed with the nanoclay. Again, the characteristic peaks associated with the loss of water in the nanoclay appear at 197, 298, and 377 °C, as was described in previous cases.

In previous works it has been possible to observe how the thermal stability of hybrids increased. Experimentally, it can be observed that the peaks of the dyes disappear when adsorbed by the nanoclay. It is considered that there are two reasons why the improvement of this property is achieved [50,51]. On the one hand, the laminar structure of the nanoclay has a barrier effect which reduces the volatility of the compounds and, on the other hand, there is a transfer of energy between the dye and the nanoclay when they are subjected to a temperature that is applied to the surface of the nanoclay [52,53]. 

### 2.5. X-ray Diffraction (XRD)

The X-ray diffraction results are shown in Figure 11. This figure shows how, after calcining the nanoclay, the intralaminar space has opened to facilitate the penetration of the dye, and also how, after the adsorption process, there is a partial recomposition of the original structure due to memory so that hydrotalcite has shape memory [24,25,26,54]. After the calcination of the nanoclay, a structural collapse produced by dehydroxylation occurs inside the layers of this mineral [29]. Between 11 and 12 degrees there is a peak that shows the described characteristic and, as in the case of calcined hydrotalcite (HC), this peak disappears due to the opening of said layers.

At this point, it is worth noting that hydrotalcite has a crystalline structure [55,56] and colorants an amorphous form. The peak of the XRD is higher when more crystalline structures are present, but this structure is totally lost when the H is calcined to form HC;it is recovered to a large extent when it is hydrated again. The experiments showed that increasing the concentration of dye adsorbed on HC causes a decrease in the XRD peak since the amount of amorphous area as compared with the crystalline area increases.

### 2.6. Fourier Transform Infrared Spectroscopy FTIR-ATR Analysis

With this technique, the primary aim is to see the effect that the calcination of H produces in some bands such as that of ${\mathrm{CO}}_{3}^{-2}$, which can be observed in the 1365 cm^−1^ band [57], and another broad strong band found at 3426 cm^−1^, attributed to stretching of the O-H bonds of water molecules and hydroxyl groups [44]. It is also important to note the thin but intense and clear bands that appear at 2850 and 2917 cm^−1^ produced by CH_2_ stretching vibrations [58]. As can be seen in Figure 12, after calcination, all these bands disappear, allowing a subsequent reconstruction in the dye adsorption phase.

Figure 13, Figure 14, Figure 15 and Figure 16 show the FTIR spectra obtained for dyes before and after adsorption on HC. In the case of Direct Blue 199 (Figure 13) and Direct Blue 71 (Figure 14), the peak at 1100 cm^−1^ is attributed to acetates, formates, propionates, and benzoates [59]. Previous work has indicated that the peaks between 1400 and 1640 cm^−1^ [55,56,57,58] correspond to benzene rings and that the peak shown at 1500 cm^−1^ is characteristic of the azo bond [60] which in Sample 12 can be seen smoothly shifted to 1520 cm^−1^. Another peak can be found at 1035 cm^−1^ which corresponds to the symmetric sulfonate vibration [48]. Many of the peaks confirm the presence of the dye with more or less intensity, or do not appear at low concentrations (Samples 2 and 3), but they are appreciated when we study them in Samples 11 and 12 when we find more peaks associated with the dye adsorbed by nanoclay (look at the peaks at 1500 and 1100 cm^−1^). In the four hybrids, an increase in the intensity of the -OH (3420–3460 cm^−1^) shoulder is observed due to the loss of water bound by hydrogen to the carbonate anions -CO_3_^2−^ in the basal space. This water is also replaced in the adsorption process [61].

The FTIR spectra of Direct Red 23 are shown in Figure 15. The peaks at 3308 and 3426 cm^−1^ correspond to the phenolic -OH, C-N stretching and water could be clearly observed [62,63,64]. Another peak is observed at 1600–1610 cm^−1^, which corresponds to the aromatic groups -C = C, while the peak at 1180–1170 cm^−1^ indicates the presence of single bonds of CO and OH [62,63,64,65,66]. The peak shown at 1480–1500 is characteristic of the azo bond [60]. Another significant peak appears at 1050 cm^−1^, which corresponds to the bonds formed by the S-O [67,68] corresponding to the sulfonate groups -SO_3_ from the NaSO_3_ group.

Studying the dye Reactive Yellow DR (Figure 16), again, the band of the phenolic -OH group, N-H stretching, and water is observed at 3446 cm^−1^ [62,63,64]. The reactive dye studied has sulfate groups that can be seen at 1110 cm^−1^ [67,68]. The peak shown at 1496 cm^−1^ is characteristic of the azo bond [60]. Other peaks are observed in Samples 4 and 9 at 1600 and 1630 cm^−1^, respectively, which correspond to the aromatic groups -C=C- [55,56,57,58], in the colorant the peak is at 1618 cm^−1^. 

Table 5 summarizes the most prominent peaks analyzed in the four hybrids. Analyzing the peaks of the OH^−^ and CO_3_^2−^ group, a semi-quantitative analysis of the area of each of these peaks can be made calculating the area under the curve. The results obtained are shown in Figure 17. This figure shows how the amount of carbonate that was originally in the hydrotalcite has been reduced as, after the calcination and reconstruction of the nanoclay, the colorant became part of the nanoclay structure, occupying a large part of this OH^−^ and CO_3_^2−^ band [24,25,26].

By analyzing the results in more depth, it is possible to extract the estimated amount of other functional groups from the dyes that have been incorporated into the structure of the hybrids. For this, the initial quantity of carbonate in the hydrotalcite is taken into consideration along with the quantity which is later shown in the hybrids. The two quantities are subtracted, one from the other, to give the number of new groups that have been incorporated. The analysis of these results is shown in Figure 17. 

### 2.7. Morphology Scanning and Transmission Electron Microscopy (SEM-TEM)

Figure 18 shows some images from the nanoclay. It is worth noting the differences between hydrotalcite before and after a calcination process was applied. Morphologically, it can be observed how, after reconstruction, the structure of the nanoclay has been recovered. This reconstruction is produced by the shape memory that the nanoclay has before and after being calcination and consequent rehydration [69].

A closer look at the morphology of the surface of the nanoadsorbent can be seen in Figure 19. Graphically, it is possible to see how the hydrotalcite lamellae are distributed composed of lamellae separated by basal or interlamellar space between which anions and water molecules are inserted. It is in this space where the adsorbed elements are deposited and where the substitution of the anions that this nanoclay possesses by default are carried out.

## 3. Discussion

The mechanism of dye adsorption and shape memory recovery of HC has been demonstrated by XRD and FTIR characterization techniques. In addition, the improvement in the thermal behavior of the hybrids obtained can be observed with respect to the dyes on their own.

Thermogravimetry (TGA) also gives results that follow similar lines. The hybrids show an almost identical behavior to that of hydrotalcite alone, so the dye adsorption has very little influence on the final TGA of the hybrid. The degradation peaks of the sample are the same as those of the nanoclay without having absorbed dyes. From the other point of view, the hybrids show better resistance to temperature than the dyes on their own. Finally, using X-ray diffraction analysis, it is possible to see how the hydrotalcite partially recovers its intralaminar space due to its shape memory. This should help the fixation of the dye within the nanoclay, and the stability of the resulting hybrid.

The XRD and FTIR characterization techniques both demonstrate the presence of the dyes in the hybrid obtained. For X-ray diffraction, it is only necessary to observe how the amorphous zone increases due to the greater presence of dye, while in the FTIR, very significant peaks of the bonds of groups such as the SO_3_ sulfonate and the amino -N=N- can be seen, associated with the adsorbed dyes and not found in the nanoclay. In the semi-quantitative analysis it can be seen how some functional groups such as CO_3_^2−^ have been replaced by the dye.

In view of the results, the effectiveness of the HC nanoclay in absorbing the reactive and direct dyes used in this study for the purification of wastewater from textile dyeing processes has been demonstrated, and the maximum level of adsorption according to Equation (1) established in this work is x = 0.5. The adsorption percentages are quite high (in all cases above 95% absorption). In addition, a good fixation of the color in the nanoclay is visually appreciated, due to its homogeneity as well as its degree of stability. The intensity of the color that has resulted from the nanoclay-dye hybrid is very low, but this is not surprising considering that the nanoclay concentration was 20 g·L^−1^ while that of the colorants was 0.05 g·L^−1^. The lowest adsorption level was found in the Direct Blue 71 dye in the lowest concentrate (Sample 3), which was the dye used with the highest molecular weight. However, no significant differences were observed due to the difference in molecular weight of the direct and reactive dyes, as can be seen in Figure 4 and Table 4 which demonstrate the good adsorption capacity of H despite the differences in the molecular weight of the dyes.

Regarding the total solar reflectance (TSR) the 4 hybrids are between 60–70% of TSR. Traditional targets exhibit a total solar reflectance of 75%, so the samples studied are very close. The previous argument is reformed in which we argued that the intensity of the color is very low, looking very white due to the white color of the nanoclay. The Reactive Yellow Drimaren less concentrated samples reach higher TSR% values than the other samples.

## 4. Materials and Methods

### 4.1. Materials

Two types of anionic dyes [33], i.e., direct and reactive [70], were used for this research. The direct dyes studied were: Direct Blue 199 CI 74.180 (phthalocyanine group with molecular weight 775.17 g/mol), Direct Red 23 CI 29160 (azo dye with molecular weight 813,72 g/mol), and Direct Blue 71 CI 34140 (azo dye with 1029.88 g/mol). The reactive dye studied was: Reactive Drimaren Yellow K-2R (azo dye with molecular weight 586 g/mol) (Figure 20).

As adsorbing substance for the dyes, hydrotalcite Mg_6_Al_2_(CO_3_)(OH)_16_·4(H_2_O) [30,36,37] was chosen, which was calcined according to Dos Santos R.M.M. [44] destroying the structure and facilitating the penetration of the dye into the nanoclay [31]. Once the nanoclay was immersed in aqueous solution, the structure reconstructed itself due to its shape memory and the colorant was incorporated into the new structure. 

### 4.2. Methods

In order to obtain the calibration line of dye solution concentration as a function of the absorbance, several samples of known concentrations were made to obtain the Lambert–Beer equation for each dye [71]. These equations allowed the dye concentration remaining in the wastewater (once the nanoclay has been applied) to be determined by the absorbance measurement. Table 6 shows the line equations and the regression (R).

One of the objectives of this work was to determine the maximum amount of dye that could be absorbed. Several solutions of dyes were made at different concentrations and the ratio of HC nanoclay/dye concentration was varied following a general full factorial design of experiments (DoE) 4^2^ (Table 7), to see what quantity of dye it could be absorbed.Table 7 shows the 16 samples and the experimental conditions. In addition, the randomization option in the dye selection was used in order to avoid experimental bias. The column “x” establishes the ratio of g·L^−1^ of the nanoclay divided by g·L^−1^ of the dye, according to the following formula:(1)$$x=\frac{\mathrm{Clay}\;g\cdot L^{-1}}{\mathrm{Dye}\;g\cdot L^{-1}}$$

Equation (1) Ratio of nanoclay/dye.

**Table 7 ijms-23-09671-t007:** Samples and their concentrations under experimental conditions according to the general full factorial DoE 4^2^.

Sample Ref.	Dye	g·L^−1^ Dye	g·L^−1^ Clay	x
1	Direct Red 23	0.05	10	200
2	Direct Blue 199	0.05	10	200
3	Direct Blue 71	0.05	10	200
4	Yellow Drimaren	0.05	10	200
5	Direct Blue 71	0.05	5	100
6	Yellow Drimaren	0.05	5	100
7	Direct Red 23	0.05	5	100
8	Direct Blue 199	0.05	5	100
9	Yellow Drimaren	1	2	2
10	Direct Red 23	1	2	2
11	Direct Blue 199	1	2	2
12	Direct Blue 71	1	2	2
13	Yellow Drimaren	1	0.5	0.5
14	Direct Red 23	1	0.5	0.5
15	Direct Blue 199	1	0.5	0.5
16	Direct Blue 71	1	0.5	0.5

Each sample solution of hybrid nanoclay-dye was stirred for 24 h, the first two hours at 1600 rpm and the remainding time at 600 rpm [72]. The next step was to filter the solution by gravity for 48 h through filter paper and measure with the spectrophotometer to calculate the concentration of dye that was not absorbed by the nanoclay [73,74]. The nanoclays with the dye were lyophilized [75,76] to completely extract the water.

A Jasco V-670 double UV-VIS/NIR spectrophotometer was used to calculate the total solar reflectance (TSR) [77]; measurements were performed in the range of 2700–190 nm with a frequency of 0.5 nm. The equipment described was equipped with a double grid monochromator. The first was used for the UV-VIS region (1200 grids/mm) in which the detectors were photomultiplier tubes, while the second was used for the infrared IR region (300 grids/mm) using a PbS detector. For both detectors, the changes that occurred in the grid were automated at the wavelength that the user wished. A deuterium lamp (190–350 nm) and a halogen lamp (330–2700 nm) were used as light sources. Reflectance factors 𝜌(λ) were applied for the hybrid pigments within the 370–740 nm range with the D65 illuminant and the CIE-1964 standard observer, in order to calculate and compare the optical properties from each one [78].

The FTIR analysis was performed with a Nicolet 6700 Spectrometer equipped with a DTGS detector. In this case, a horizontal mono-rebound attenuated total reflection (FTIR-ATR) using a prism of ZnSe was used. A pressure control accessory was used to equalize the pressure in each sample. The average spectra were obtained after 64 scans with a resolution of 4 cm^−1^ by subtracting the background signals obtained.

A TGA analysis was also carried out to compare possible variations in the degradation peaks of the dyes [79,80,81]. To perform the characterization of the thermal properties of the hybrids, in this study, a thermogravimetric analyzer TGA/SDTA 851 (Mettler-Toledo Inc., Columbus, OH, USA) was used. For this, the conditions of the experiment that were established included an increase in temperature of 5 °C every minute within the range from 20 to 900 °C, and the oxidizing medium used was N_2_/O_2_ (4:1). 

Finally, an X-ray diffraction (XRD) test [82,83] was carried out to verify that the nanoclay recovered its original shape before it was calcined. An XRD bruker D8-Advance equipment (Bruker, Billerica, MA, USA) with a Göebel mirror (power 3000 W, voltage 20–60 kV, and current 5–80 mA) was used. Measurements were taken in an oxidant atmosphere at an angular speed of 1°/min, STEP 0.05°, and an angular scan of 2.7–70°. The XRD patterns allowed us to check the memory form and reconstruction of the hydrotalcite structure after the calcination process and were also used to find variations in the basal space on the nanoclay structure due to the interactions with the selected dyes.

ANOVA was used as the statistical method to determine whether the dye adsorptions (mean values) of two or more groups were different, using the dye structure and the dye/clay concentrations as factors for the analysis. The ANOVA studies were carried out with 95% confidence level. The order of use of the dyes was not correlative in order to avoid experimental bias.

## Figures and Tables

**Figure 1 ijms-23-09671-f001:**
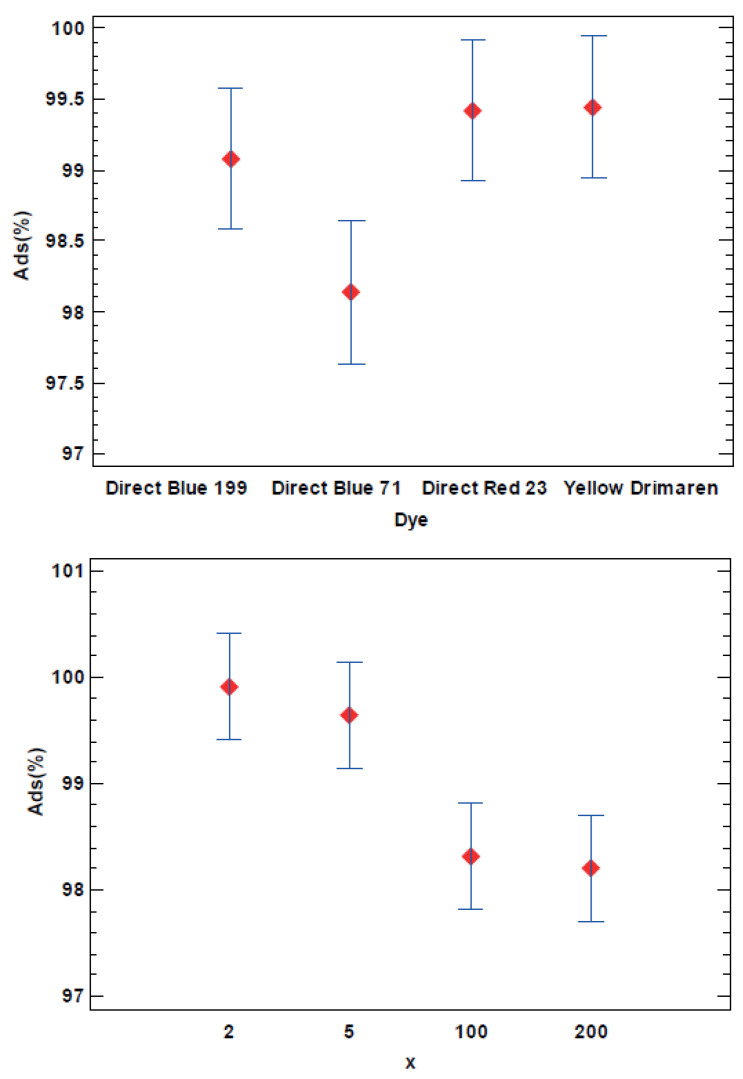
LSD Fisher means plot (95% confidence intervals) from Ads(%) as the response variable.

**Figure 2 ijms-23-09671-f002:**
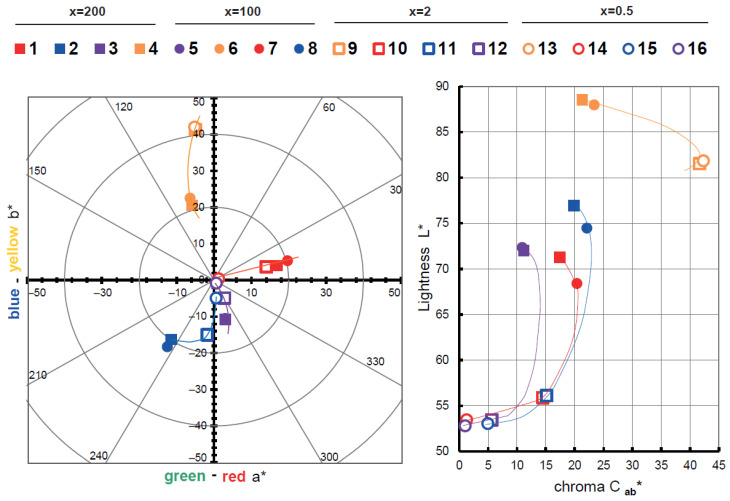
Graphic CIELAB plots for hybrid pigments synthetized under 1–18 conditions using the D65 illuminant and the CIE-1931 XYZ standard observer. Left, CIE-a* b* color diagram; right, CIE-Cab* L* color chart.

**Figure 3 ijms-23-09671-f003:**
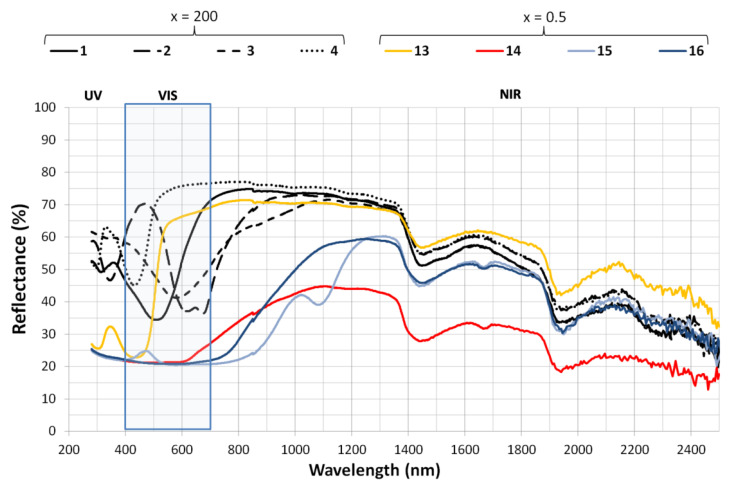
TSR (%) for each hybrid at maximum and minimum dye loading.

**Figure 4 ijms-23-09671-f004:**
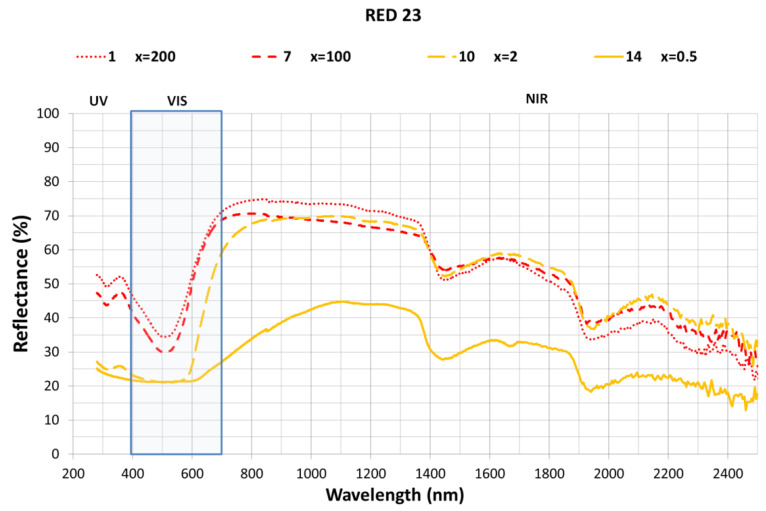
TSR (%) for RED 23 dye hybrid.

**Figure 5 ijms-23-09671-f005:**
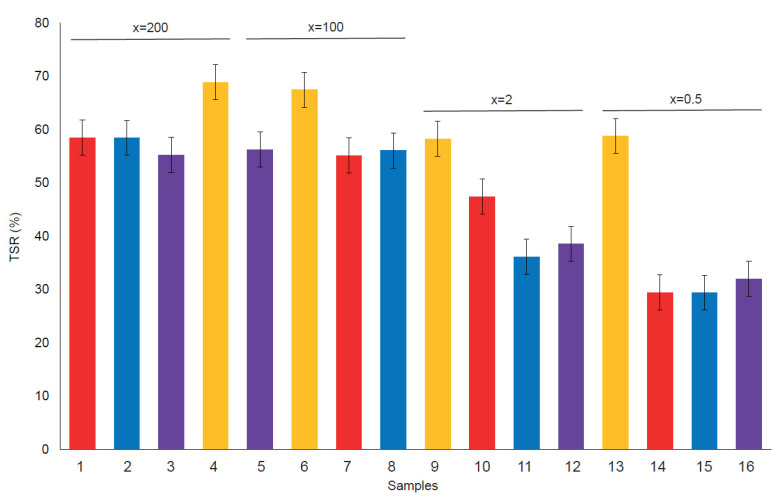
Total Solar Reflectance (TSR (%)) for each sample.

**Figure 6 ijms-23-09671-f006:**
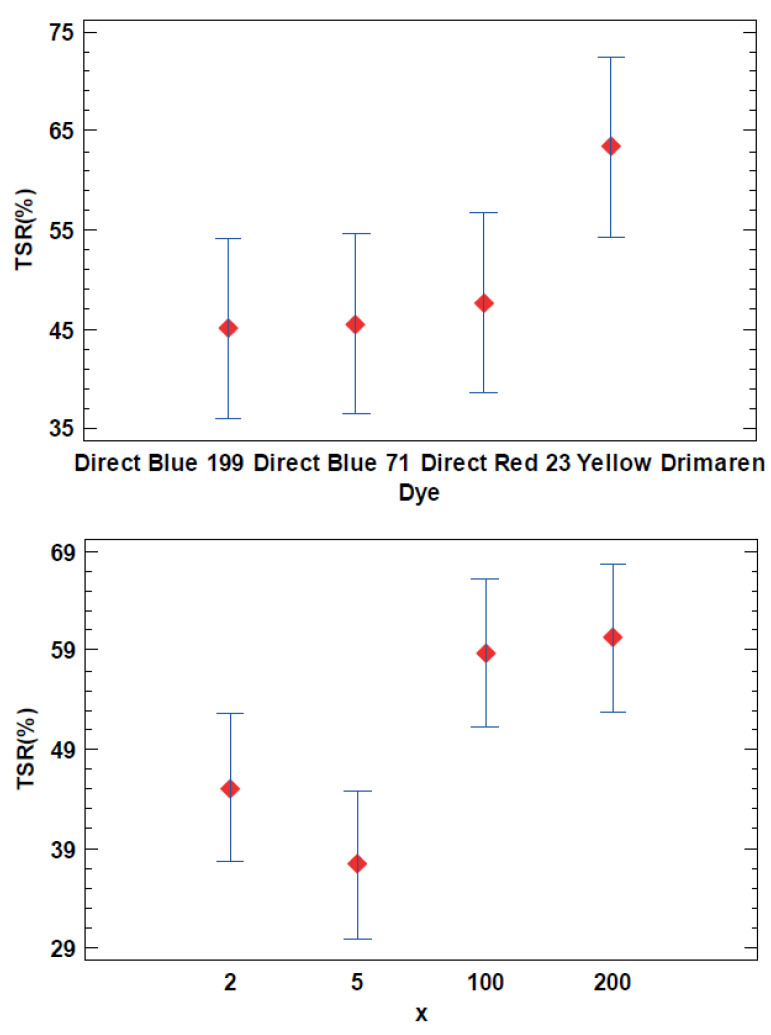
LSD Fisher means plot (95% confidence intervals) from TSR (%) as the response variable.

**Figure 7 ijms-23-09671-f007:**
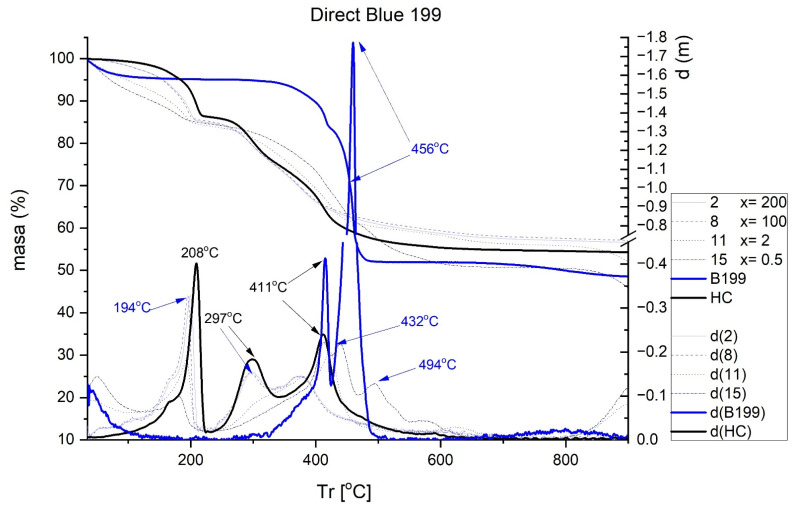
TGA and DTGA of Direct Blue 199 free and adsorbed.

**Figure 8 ijms-23-09671-f008:**
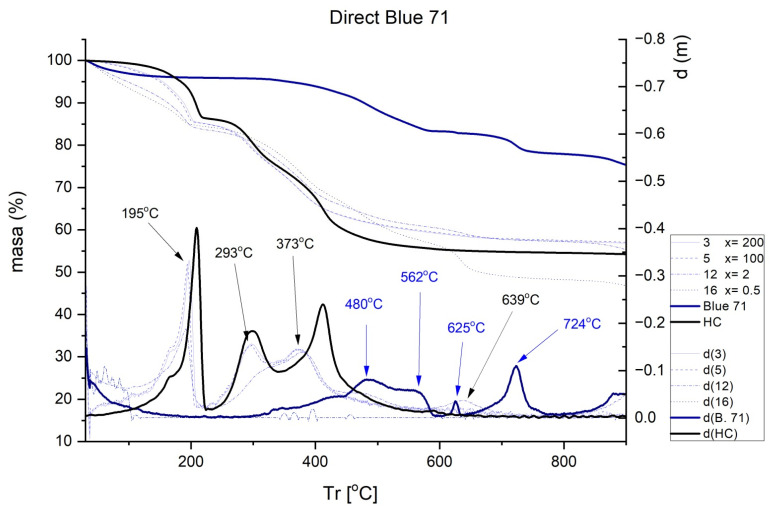
TGA and DTGA of Direct Blue 71 free and adsorbed.

**Figure 9 ijms-23-09671-f009:**
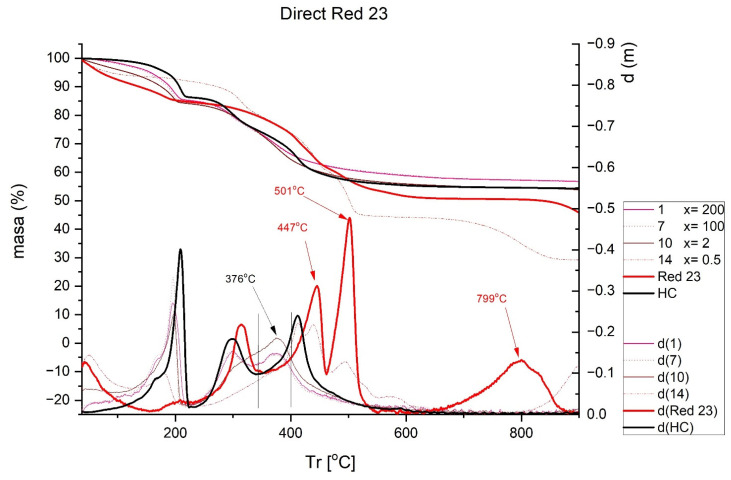
TGA and DTGA of Direct Red 23 free and adsorbed.

**Figure 10 ijms-23-09671-f010:**
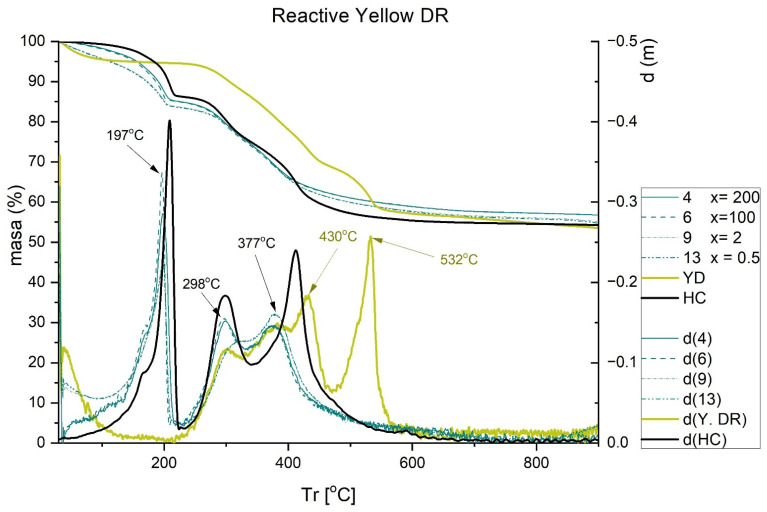
TGA and DTGA of Reactive Yellow DR free and adsorbed.

**Figure 11 ijms-23-09671-f011:**
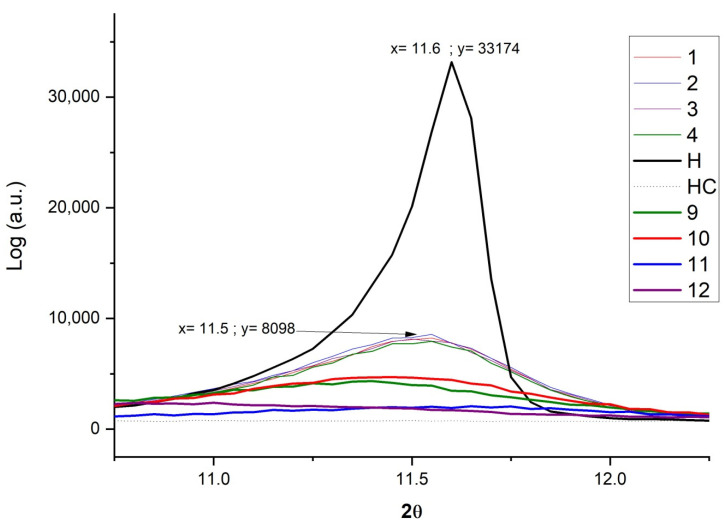
XRD for hydrotalcite, hydrotalcite calcinated, Samples 1–4 and Samples 9–12.

**Figure 12 ijms-23-09671-f012:**
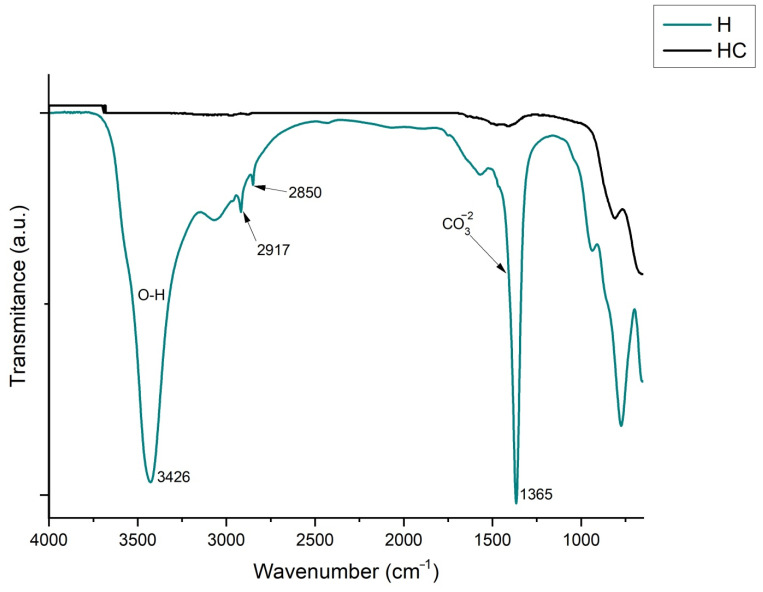
FTIR of uncalcined hydrotalcite (H) and calcined hydrotalcite (HC).

**Figure 13 ijms-23-09671-f013:**
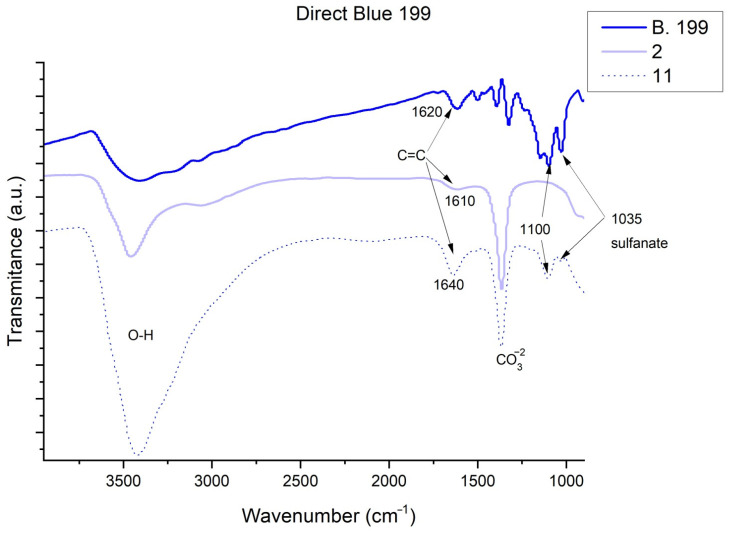
FTIR of Direct Blue 199, Sample 2 and Sample 11.

**Figure 14 ijms-23-09671-f014:**
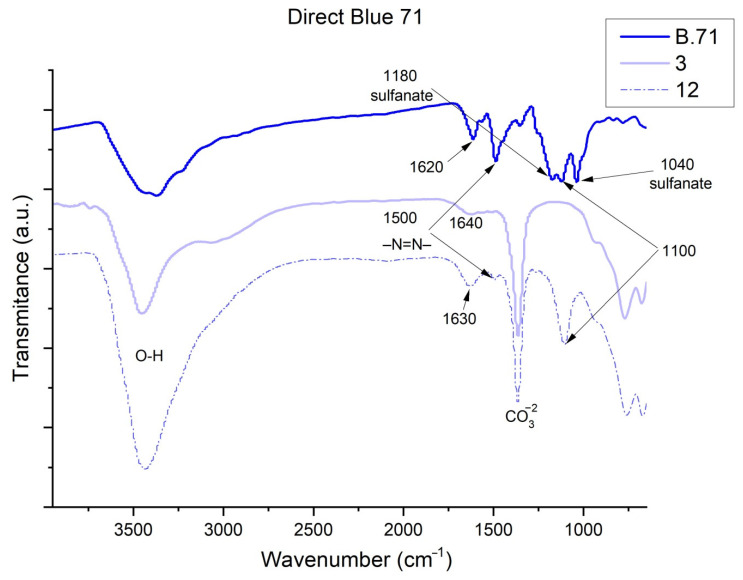
FTIR of Direct Blue 71, Sample 3 and Sample 12.

**Figure 15 ijms-23-09671-f015:**
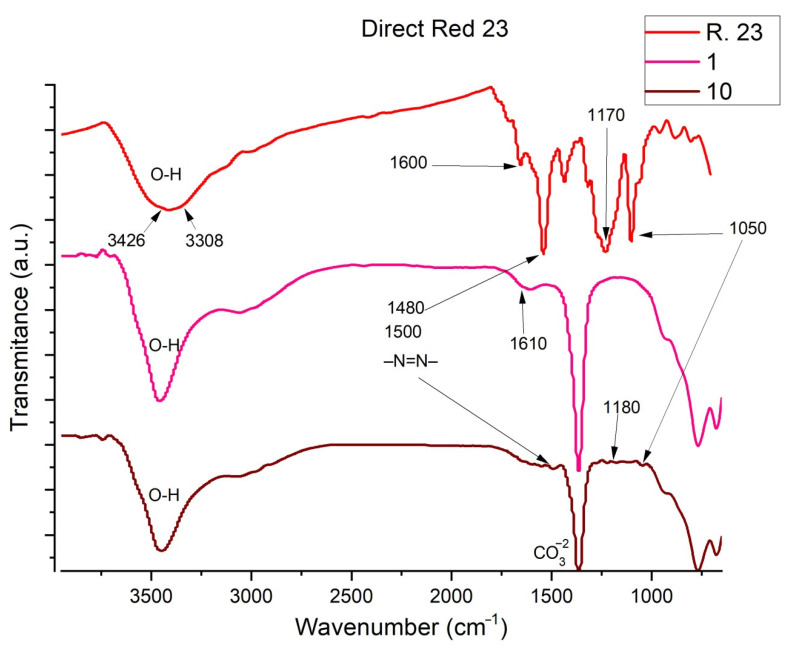
FTIR of Direct Red 23, Sample 1 and Sample 10.

**Figure 16 ijms-23-09671-f016:**
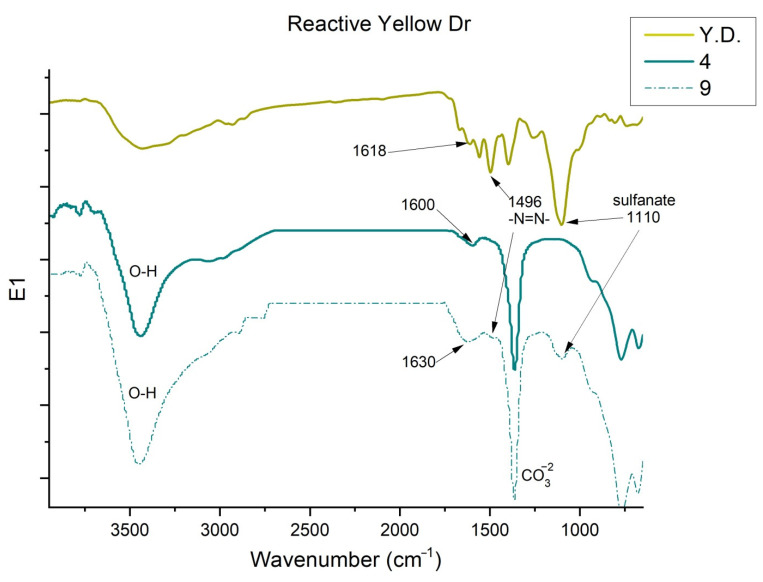
FTIR of Yellow DR, Sample 4 and Sample 9.

**Figure 17 ijms-23-09671-f017:**
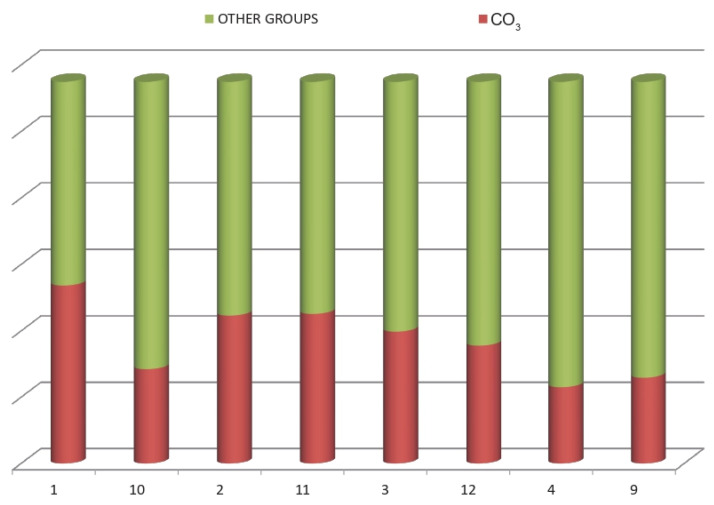
Semi-quantitative analysis and incorporation of new functional groups.

**Figure 18 ijms-23-09671-f018:**
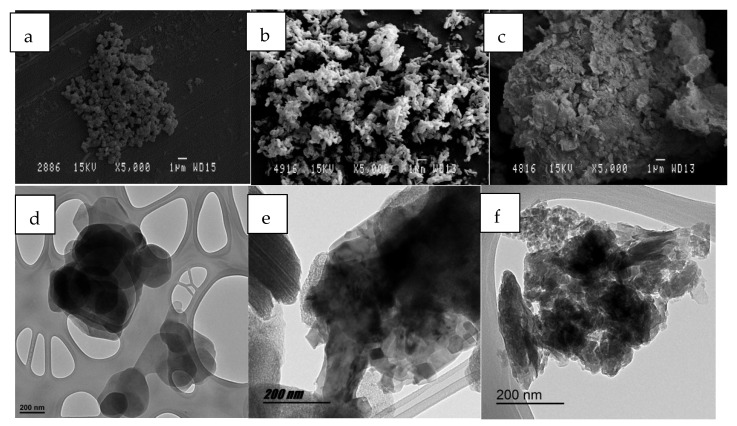
SEM micrographies of different HC samples: (**a**) HC original; (**b**) HC calcinated; (**c**) HC reconstructed. TEM micrographies of different HC samples: (**d**) HC original; (**e**) HC calcinated; (**f**) HC reconstructed.

**Figure 19 ijms-23-09671-f019:**
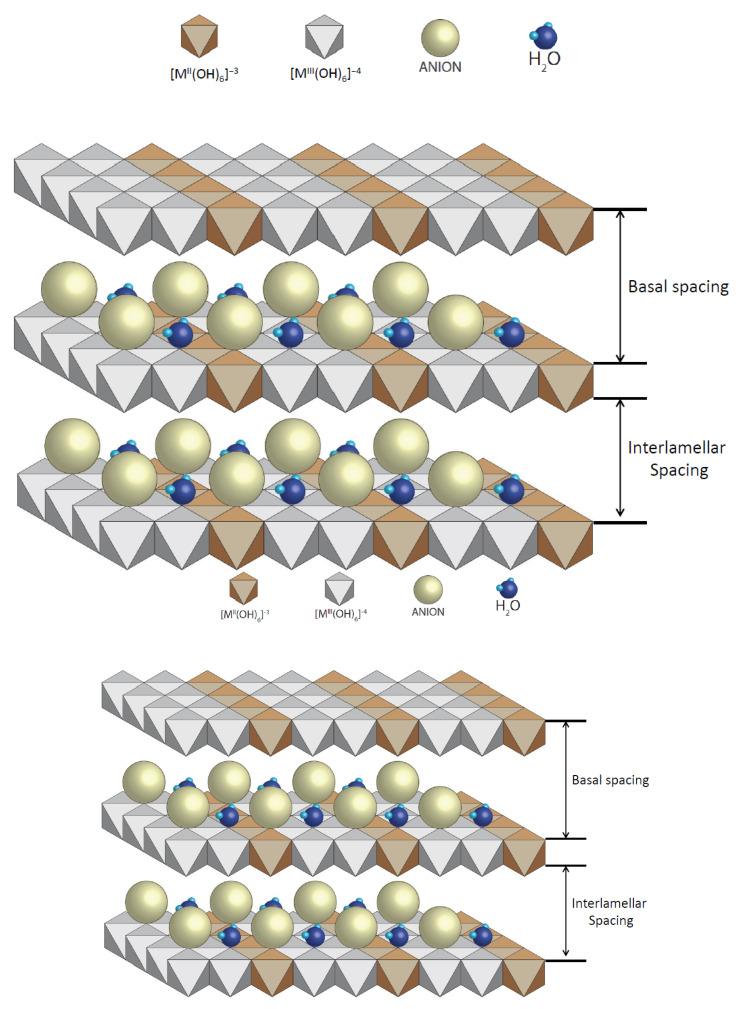
Basic structure of hydrotalcite [34].

**Figure 20 ijms-23-09671-f020:**
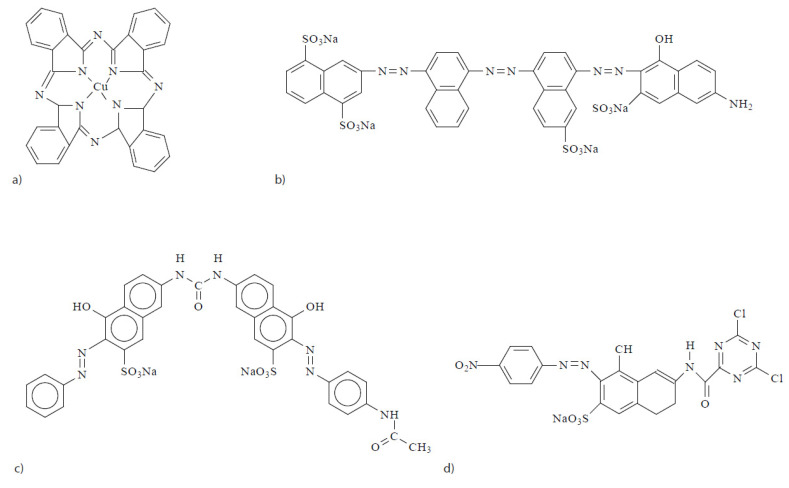
(**a**) Direct Blue 199; (**b**) Direct Blue 71; (**c**) Direct Red 23; (**d**) Reactive Drimaren Yellow.

**Table 1 ijms-23-09671-t001:** Differences in concentrations after HC adsorption.

Sample Ref.	Dye	Initial Conc. g·L^−1^	Final Conc. g·L^−1^	Absorbance	Ads (%)
1	Direct Red 23	0.05	4.89 × 10^−4^	0.002	99.22
2	Direct Blue 199	0.05	7.80 × 10^−4^	0.002	98.43
3	Direct Blue 71	0.05	1.77 × 10^−3^	0.007	96.45
4	Yellow Drimaren	0.05	5.42 × 10^−4^	0.006	98.91
5	Direct Blue 71	0.05	1.60 × 10^−3^	0.004	96.81
6	Yellow Drimaren	0.05	3.41 × 10^−4^	0.003	99.32
7	Direct Red 23	0.05	5.47 × 10^−4^	0.004	98.91
8	Direct Blue 199	0.05	8.72 × 10^−4^	0.004	98.26
9	Yellow Drimaren	1	7.43 × 10^−4^	0.009	99.93
10	Direct Red 23	1	5.76 × 10^−4^	0.005	99.94
11	Direct Blue 199	1	8.26 × 10^−4^	0.003	99.92
12	Direct Blue 71	1	1.71 × 10^−3^	0.006	99.83
13	Yellow Drimaren	1	3.89 × 10^−3^	0.056	99.61
14	Direct Red 23	1	1.94 × 10^−3^	0.052	99.81
15	Direct Blue 199	1	2.85 × 10^−3^	0.047	99.72
16	Direct Blue 71	1	5.34 × 10^−3^	0.068	99.47

**Table 2 ijms-23-09671-t002:** Variance analysis for Ads(%) from the 4^2^ DoE.

Source	Sum of Squares	f.d.	Medium Square	F-Ratio	*p*-Value
A:Dye	2.59932	1	2.59932	5.06	0.0482
B:x	2.14673	1	2.14673	4.18	0.0681
AA	2.6937	1	2.6937	5.24	0.045
AB	0.0905298	1	0.0905298	0.18	0.6835
BB	1.40323	1	1.40323	2.73	0.1294
Total Error	5.13621	10	0.513621		
Total (corr.)	17.3029	15			

R-squared = 70.3159%.

**Table 3 ijms-23-09671-t003:** TSR Values.

Dye	Sample Ref.	TSR %
Direct Red 23	1	58.50
Direct Blue 199	2	58.45
Direct Blue 71	3	55.27
Yellow Drimaren	4	63.45
Direct Blue 71	5	56.27
Yellow Drimaren	6	67.49
Direct Red 23	7	55.14
Direct Blue 199	8	56.08
Yellow Drimaren	9	58.30
Direct Red 23	10	47.44
Direct Blue 199	11	36.14
Direct Blue 71	12	38.55
Yellow Drimaren	13	58.80
Direct Red 23	14	29.47
Direct Blue 199	15	29.42
Direct Blue 71	16	32.00

**Table 4 ijms-23-09671-t004:** Analisys of variance for TSR values.

Source	Sum of Squares	f.d.	Medium Square	F-Ratio	*p*_Value
A:Dye	227.623	1	227.623	6.48	0.029
B:x	478.524	1	478.524	13.63	0.0042
AA	418.847	1	418.847	11.93	0.0062
AB	43.4298	1	43.4298	1.24	0.292
BB	178.111	1	178.111	5.07	0.048
Error total	351.007	10	35.1007		
Total (corr.)	2580.2	15			
R-squared = 86.3961%					

**Table 5 ijms-23-09671-t005:** A summary of the IR peak positions.

	H	B199	2	11	B71	3	12	R23	1	10	YD	4	9
-OH	3426	-	3460	3420	3426	3460	3440	3308	3460	3450	3438	3442	3450
-CO_3_^2−^	1365	-	1370	1370	-	1370	1370	-	1370	1370	-	1370	1370
-N=N-	-	1500	-	1520	1500	-	1500	1480	-	1500	1496	-	1496
-SO_3_	-	1035	-	1035	1180	-	1100	1050	-	1050	1110	-	1110
-C=C-	-	1620	1610	1640	1620	1640	1630	1600	1610	-	1618	1600	1630

**Table 6 ijms-23-09671-t006:** Lambert–Beer line equations and R^2.^

Dye	Equation	R^2^
Direct Blue 199 (B199)	y = 21.784 x − 0.015	0.9982
Reactive Yellow (YD)	y= 14.943 x − 0.0021	0.9993
Direct Red 23 (R23)	y= 34.357 x − 0.0148	0.9991
Direct Blue 71 (B71)	y= 17.09 x − 0.0233	0.9987

## Data Availability

Not applicable.

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
