# Peer review of "The Optimal Concentration of Nanoclay Hydrotalcite for Recovery of Reactive and Direct Textile Colorants"

_ijms, 2022, doi:10.3390/ijms23179671_

Round 1
Reviewer 1 Report
This is an interesting piece of work. The methodology is sound and the results clearly presented.
However, the sections must be re-structured, as materials and methods appears as section 4 , results as section 2 and discussion as section 3. It should be section 2 materials and methods, section 3 results and section 4 discussion.
In terms of English style, there are minor spelling mistakes. Please review the document.
Regarding the results related to TGA, Figures 7 and 8 seem to be redundant and not necessary. Instead, I would add the TGA plots of HC and dye to Figures 9 to 12 and remove 7 and 8.
Regarding the analysis of the hydrotalcite before and after calcination, the presentation of results should be consistent. SEM and TEM images are included in the materials section, while the FTIR plots are in the results section. I suggest that the authors present both sets of results in the same section.
Author Response
Dear Reviewer. The revision on your part is correct, we will respond to each of your comments and you will find attached the document with the modifications.
- However, the sections must be re-structured, as materials and methods appears as section 4 , results as section 2 and discussion as section 3. It should be section 2 materials and methods, section 3 results and section 4 discussion.
Although in many magazines the order of the structure is as you indicate in this magazine we have followed the instructions in this URL https://www.mdpi.com/journal/ijms/instructions#manuscript and in the template, the order of the structure indicated in these instructions (2.results, 3.Discussion, 4.materials and methods). Therefore the structure is correct and should not be modified as you suggest.
- In terms of English style, there are minor spelling mistakes. Please review the document.
The English has been proofread by a native translator to ensure that there are no mistakes.
- Regarding the results related to TGA, Figures 7 and 8 seem to be redundant and not necessary. Instead, I would add the TGA plots of HC and dye to Figures 9 to 12 and remove 7 and 8.
We have removed the old figures 7 and 8 and the HC curves have been integrated into the new graphs now numbered 7 to 10.
- Regarding the analysis of the hydrotalcite before and after calcination, the presentation of results should be consistent. SEM and TEM images are included in the materials section, while the FTIR plots are in the results section. I suggest that the authors present both sets of results in the same section.
Following your recommendations the SEM-TEM analysis has been placed in section 2.7 on page 18.
Reviewer 2 Report
The authors investigated toxic dye removal using nano clay hydrotalcite adsorbent to safeguard public health. The appropriate adsorbent and methods are always welcome in removing the toxic pollutants based on the selectivity and sensitivity parameters. Some important points are important to be addressed before going to consider the possible publication in this journal.
-The English language needs to check carefully in the revision stage because of many careless mistakes in many positions.
-The Figures quality needs to be improved in the revision stage.
-References: There are many references that are not adjacent to this study. The authors need to take notes in the revision stage and cite relevant references including high-impact journals to make the manuscript to a broad range of readers.
-Abstract: This section needs to be improved by presenting novel findings such as investigating the divers metal ions effect, and the main experimental works in the revision stage.
-Introduction: No doubt, high adsorption with selectivity is the key points for materials potentiality of dye removal. Composite materials are growing attention for diverse dyes removal based on their specific functionality and surface area as reported by Awual group according to ScienceDirect. The authors need to indicate such points for a broad range of readers. Moreover, the authors need to cite high-impact articles to make the manuscript high-level. The following specific articles may take be noted in the revision stage of Journal of Molecular Liquids, 323 (2021) 114587; Journal of Molecular Liquids 328 (2021) 115468.
-Scientists considered the removal technology based on selectivity, sensitivity, cost-effectiveness, and so on. The authors need to indicate such a point in the revised manuscript.
-The materials characterization is not accurate. The surface morphology needs to be added instead and then compared with reported materials including Journal of Molecular Liquids 329 (2021) 115541.
-Can the material be effective to use dye removal from the real waste samples? The authors need to write such sentences in the revision stage.
I would like to see the revised manuscript.
Author Response
Dear Reviewer. The revision on your part is correct, we will respond to each of your comments and you will find attached the document with the modifications.
- The English language needs to check carefully in the revision stage because of many careless mistakes in many positions.
The English has been proofread by a native translator to ensure that there are no mistakes.
- The Figures quality needs to be improved in the revision stage.
The images have been revised and improved as much as possible. In addition, after consultation with the editor, we have attached all the images in vector format so that in the final assembly of the document they can be put in the size and quality you want.
- References: There are many references that are not adjacent to this study. The authors need to take notes in the revision stage and cite relevant references including high-impact journals to make the manuscript to a broad range of readers.
Very good recommendation, we appreciate it because it is certainly a great way to increase the value of the manuscript. Relevant new references from high impact journals have been added in response to your input. Some examples of these new references and the journal impact factor of these journals are shown below.
Aregay, G. G.; Jawad, A.; Du, Y.; Shahzad, A.; Chen, Z. Efficient and Selective Removal of Chromium (VI) by Sulfide Assembled Hydrotalcite Compounds through Concurrent Reduction and Adsorption Processes. J. Mol. Liq. 2019, 294, 111532.
Yadav, M. K.; Gupta, A. K.; Ghosal, P. S.; Mukherjee, A. PH Mediated Facile Preparation of Hydrotalcite Based Adsorbent for Enhanced Arsenite and Arsenate Removal: Insights on Physicochemical Properties and Adsorption Mechanism. J. Mol. Liq. 2017, 240, 240–252.
JOURNAL OF MOLECULAR LIQUIDS JIC 6.633
Orthman, J.; Zhu, H. Y.; Lu, G. Q. Use of Anion Clay Hydrotalcite to Remove Coloured Organics from Aqueous Solutions. Sep. Purif. Technol. 2003, 31 (1), 53–59. https://doi.org/10.1016/S1383-5866(02)00158-2.
SEPARATION AND PURIFICATION TECHNOLOGY JIC 9.136
Bascialla, G.; Regazzoni, A. E. Immobilization of Anionic Dyes by Intercalation into Hydrotalcite. Colloids Surfaces A Physicochem. Eng. Asp. 2008, 328 (1–3), 34–39.
SURFACES A-PHYSICOCHEMICAL AND ENGINEERING ASPECTS JIC 5.518
Bouraada, M.; Lafjah, M.; Ouali, M. S.; de Menorval, L. C. Basic Dye Removal from Aqueous Solutions by Dodecylsulfate-and Dodecyl Benzene Sulfonate-Intercalated Hydrotalcite. J. Hazard. Mater. 2008, 153 (3), 911–918.
JOURNAL OF HAZARDOUS MATERIALS JIC 14.224
Lazaridis, N. K.; Karapantsios, T. D.; Georgantas, D. Kinetic Analysis for the Removal of a Reactive Dye from Aqueous Solution onto Hydrotalcite by Adsorption. Water Res. 2003, 37 (12), 3023–3033.
WATER RESEARCH JIC 13.4
Yu, L.; Deng, J.; Wang, H.; Liu, J.; Zhang, Y. Improved Salts Transportation of a Positively Charged Loose Nanofiltration Membrane by Introduction of Poly (Ionic Liquid) Functionalized Hydrotalcite Nanosheets. ACS Sustain. Chem. Eng. 2016, 4 (6), 3292–3304.
ACS Sustainable Chemistry & Engineering JIC 9.224
Biglari Quchan Atigh, Z.; Heidari, A.; Karimi, A.; Pezhman, M. A.; Asgari Lajayer, B.; Lima, E. C. Purification and Economic Analysis of Nanoclay from Bentonite. Environ. Sci. Pollut. Res. 2021, 28 (11), 13690–13696.
ENVIRONMENTAL SCIENCE AND POLLUTION RESEARCH JIC 5.19
- Abstract: This section needs to be improved by presenting novel findings such as investigating the divers metal ions effect, and the main experimental works.
We have added to the abstract: As a new finding it has been possible to analyze and quantify the amount of metal ions substituted by anionic dyes when adsorbed, determining the optimal amount of clay to be used to adsorb all the dye.
- Introduction: No doubt, high adsorption with selectivity is the key points for materials potentiality of dye removal. Composite materials are growing attention for diverse dyes removal based on their specific functionality and surface area as reported by Awual group according to ScienceDirect. The authors need to indicate such points for a broad range of readers. Moreover, the authors need to cite high-impact articles to make the manuscript high-level. The following specific articles may take be noted in the revision stage of Journal of Molecular Liquids, 323 (2021) 114587; Journal of Molecular Liquids 328 (2021) 115468.
See in page 2: The Awual group has been reporting a growing trend for the use of composite materials for the removal of different dyes, based on the specific surface area of the adsorbing agent and functionality [19]–[21].
The articles “Journal of Molecular Liquids, 323 (2021) 114587; Journal of Molecular Liquids 328 (2021) 115468” are very interesting for our study and we have added them to the bibliography with reference to basic (cationic) dyes.
- Scientists considered the removal technology based on selectivity, sensitivity, cost-effectiveness, and so on. The authors need to indicate such a point in the revised manuscript.
In this case we can consider the elimination of pollutants in effluents because of their cost-effectiveness, as clays are abundant and cheap. On page 2 of the manuscript the following text and bibliographical references have been added: “In addition, nanoclays are an abundant and low-cost material, which makes them very cost-effective considering the cost-effectiveness ratio.”
- The materials characterization is not accurate. The surface morphology needs to be added instead and then compared with reported materials including Journal of Molecular Liquids 329 (2021) 115541.
The following paragraphs and new figure 19 have been added on pages 18 and 19:
Figure 18 shows some images from the nanoclay. It is worth noting the differences between hydrotalcite before and after a calcination process was applied. Morphologically, it can be observed how after reconstruction, the structure of the clay has been recovered. This reconstruction is produced by the shape memory that the clay has before and after being calcined and consequent rehydration.
A closer look at the morphology of the surface of the nanoadsorbent can be seen in figure 19. Graphically, it is possible to see how the hydrotalcite lamellae are distributed composed of lamellae separated by basal or interlamellar space between which anions and water molecules are inserted. It is in this space where the adsorbed elements will be deposited and where the substitution of the anions that this clay possesses by default will be carried out.
In addition, a description of the amorphous and crystalline zones can be found on page 13, based on the information obtained from the XRD analysis.
- Can the material be effective to use dye removal from the real waste samples? The authors need to write such sentences in the revision stage.
In response to your comments, we have added the following on page 3: “Prior to these analyses, dye adsorption tests were carried out on several real wastewaters collected from a textile dyeing industry, thus confirming the adsorption capacity and making the results of this work extrapolate to the real industry.”